# Flexible Tellurium-Based Electrode for High-Performance Lithium-Tellurium Battery

**DOI:** 10.3390/nano11112903

**Published:** 2021-10-29

**Authors:** Yan Li, Ye Zhang

**Affiliations:** 1School of Resource & Environment and Safety Engineering, University of South China, Hengyang 421001, China; yanli_usc@163.com; 2School of Chemistry and Chemical Engineering, University of South China, Hengyang 421001, China

**Keywords:** tellurium nanotubes, nanofibrillated cellulose, flexible electrode, Li-Te battery

## Abstract

Low-dimensional nanomaterials have attracted considerable attention for next-generation flexible energy devices owing to their excellent electrochemical properties and superior flexibility. Herein, uniform Tellurium nanotubes (Te NTs) were prepared through a facile hydrothermal method, and then a flexible and freestanding electrode was fabricated with Te NTs as active materials and a small amount of nanofibrillated celluloses (NFCs) as a flexible matrix through a vacuum filtration method without adding extra conductive carbon or a binder. The resulting Te-based electrode exhibits a high volumetric capacity of 1512 mAh cm^−3^ at 200 mA g^−1^, and delivers admirable cyclic stability (capacity retention of 104% over 300 cycles) and excellent rate performance (833 mAh cm^−3^ at 1000 mA g^−1^), which benefits from the unique structure and intrinsically superior conductivity of Te NTs. After bending 50 times, the Te-based electrode delivers a desirable volumetric capacity of 1117 mAh cm^−3^, and remains 93% of initial capacity after 100 cycles. The results imply that the Te-based electrode exhibits excellent electrochemical properties and superior flexibility simultaneously, which can serve as a potential candidate for the flexible lithium batteries.

## 1. Introduction

In today’s high-tech age, intelligent and portable devices are expected to bring great changes to our life, and drive a new consumption tide of flexible and wearable electronic products [1]. The development of advanced energy storage devices with high energy density and superior flexibility is the foundation of constructing flexible electronics, which has become a main challenge in realizing the practical application of flexible devices. Lithium batteries show great promise in the application field of flexible power sources due to their attractive properties such as having high energy density, long cycle life, and so forth [2,3,4]. However, the traditional commercial lithium batteries are inflexible and unbending, which would lead to a sharp decline in electrochemical performance or even serious security problems when applied to flexible devices. At present, many approaches have been proposed to solve these problems. A well-known strategy to solve the problem is using a flexible matrix instead of the metal foil as a current collector. Cellulose is considered one of the largest natural biological materials in the nature. Nanofibrillated celluloses (NFCs) degraded from cellulose show many excellent characteristic, such as high specific surface area, superior flexibility, good biocompatibility, and degradability, thus becoming an ideal matrix for flexible electrode [5,6]. Feng et al. [7] have reported a composite electrode with NFCs as flexible matrix, which demonstrated good flexibility and electrochemical performance. However, the further enhancement of the flexibility of such composite electrode will be limited by the rigid active materials to some extent. Besides, the addition of a large proportion of inactive substances including flexible matrix, binder, and conductive additive, will lead to the reduction of the overall energy density of the flexible battery. The demand for flexible and portable electronic products has prompted great efforts in searching for novel electrode materials, and the exploitation of active materials which exhibiting both intrinsic flexibility and excellent electrochemical properties has become a key challenge to the application of fully flexible electronic devices.

Recently, low-dimensional nanomaterials with a particular structure have been widely applied in sensors, optoelectronics, biological medicine, and energy devices [8,9]. They have also attracted considerable attention for the applications of flexible lithium batteries owing to their attractive properties such as high specific surface area, enhanced conductivity, remarkable flexibility, and high reversible lithium storage capacity [10]. For instance, Ning et al. [11] fabricated a flexible graphene paper using a vermiculite-templated chemical vapor deposition process, and the flexible anode exhibited not only outstanding mechanical flexibility but also good electrochemical properties, especially high electrical conductivities. Chen et al. [12] demonstrated a flexible BP-G hybrid paper, in which black phosphorus nanosheets are composited with few-layer graphene as a high-performance flexible electrode. As a member of the chalcogen family, Te exhibits a remarkable theoretical volumetric capacity of 2621 mAh cm^−3^, which is propitious for the battery packing and the rational design of electric devices. In addition, Te possesses the strongest metal quality and the highest electronic conductivity (2.5 S cm^−1^) among all non-metallic elements, thus resulting in a better rate capability in lithium batteries. Various Te-bsed materials have been fabricated and demonstrated in high-performance Li-Te batteries according to previous studies [13,14,15]. However, most of the Te-based electrodes use carbon hosts or graphene to alleviate the large volume change of Te, and there are few attempts to apply Te as a flexible electrode for Li-Te battery. In this contribution, Uniform Te NTs are successfully prepared through a facile hydrothermal method. Then, a flexible and freestanding tellurium-based (Te-based) electrode is fabricated from Te NTs and NFCs without extra conductive carbon and binder via a facile vacuum filtration method. The particular structure of one-dimensional equips Te NTs with large specific surface area, high flexibility, and good film-formation ability. The attractive structural features will significantly increase the active sites for electrochemical reaction, and favor the rapid ion diffusion at the same time. What is more, a small number of NFCs were used to form the freestanding electrode, which can avoid using metal as fluid collection and greatly improve the flexibility of the electrode. With these merits, the Te-based electrode is expected to be a promising candidate for flexible lithium batteries.

## 2. Materials and Methods

### 2.1. Synthesis of Te NTs

Typically, 115 mg Na_2_TeO_3_ (≥99.9%, Sigma-Aldrich, Inc. Shanghai, China) and 1.2 g PVP (Mw~30,000, Sigma-Aldrich, Inc.) were added into 90 mL of deionized water and were stirred for 0.5 h to form a well-dissolved solution. Then, 12 mL NH_3_·H_2_O (≥25%, Aladdin Co. Shanghai, China) and 6 mL N_2_H_4_·H_2_O (≥85%, Aladdin Co.) were emptied into the above solution, which was stirred for 10 min. Afterwards, the resultant solution was transferred into a 125 mL Teflon-lined autoclave and heated in an oven at 160 °C for 24 h. The prepared sample was collected after cooling to room temperature and was then cleaned through washing and centrifuge three times. The resulting product with a silver-gray color was obtained after overnight vacuum drying.

### 2.2. Fabrication of the Flexible and Freestanding Te-Based Electrode

Figure 1 illustrates the flow diagram of the whole synthesis for the flexible Te-based electrode. A proper amount of Te NTs and NFCs (solid content = 5%, NingBo EneRol Nanotechnologies, Inc. Ningbo, China) were separately added into 50 mL ethanol and then ultrasound dispersed evenly. The mass ratio of Te NTs and NFCs are 8:1. The obtained NFCs suspension and Te NTs dispersion were successively filtered over a PTFE membrane (0.22 µm pore size, Tianjin Jinteng Experiment Equipment Co., Ltd. Tianjin, China). The resulting filter cake was dried for 12 h, and then cut to a thin film electrode with a 10 mm diameter. The coated thickness of the electrode is about 4~5 µm. The average weight of the active materials of each thin film electrode was about 2 mg. 

### 2.3. Materials Characterizations

The morphologies and structures of the Te NTs and the Te-based electrode were studied by scanning electron microscopy (SEM, Hitachi-SU8010, Hitachi, Tokyo, Japan) and transmission electron microscopy (TEM, FEI Tecnai G2 F30, FEI Company, Eindhoven, The Netherlands). X-ray diffraction (XRD) analysis was carried out using the X’Pert-Pro MPD diffractometer (Cu K-α radiation, λ = 1.7903 Å). Raman spectra were recorded with a Renishaw spectrometer using a silicon wafer at 633 cm^−1^ as the reference to calibrate the spectrometer.

### 2.4. Electrochemical Measurement

The electrochemical behaviors of the resulting Te-based electrode were investigated using a CHI660e electrochemical workstation (Chenhua Instruments Co. Ltd. Shanghai, China) and a Neware battery testing system (Neware Technology Co.,Ltd. Shenzhen, China) by assembling into a CR2032 coin-type cell (Neware Technology Co.,Ltd. Shenzhen, China). Li metal foil (15.6 mm in diameter and 0.45 mm thick), microporous polypropylene film and the liquid organic electrolytes [1 M LiPF_6_ in EC/DMC/EMC (1:1:1 by volume)] purchased from DodoChem were used as the counter electrodes, separator and electrolyte, respectively. The charge–discharge test was performed in the potential range of 0.01–4 V at 25 °C though the capacity of the voltage below 1 V will be hardly exploitable by practical electronic devices. An EIS test was performed with the frequency ranges from 0.01 to 10^5^ Hz. A cyclic voltammetry (CV) measurement was carried out at a scan rate of 0.1 mV s^−1^. To further evaluate the feasibility of the Te-based electrode as a flexible electrode for lithium batteries, a cyclical bending test and the corresponding electrochemical test of the Te-based electrode are also performed.

## 3. Results and Discussions

Figure 2a,b shows the highly anisotropic structure of Te, the covalently bonded atoms form the unique helical chains which orient along the *c*-axis and band together through Van Der Waals interactions [16,17]. The SEM image shows that the prepared Te NTs have regular long tubular shapes with about 120 nm in diameter (Figure 2c). Figure 2d,e show the TEM images of the typical Te NTs; the hollow structure of the Te NTs can be clearly observed, and the inside diameter and wall thickness of Te NTs are about 60 nm and 30 nm, respectively. As illustrated in the HRTEM image (Figure 2f), the lattice space of the Te NTs is 0.386 nm, corresponding to the (100) planes of Te (PDF No# 36-1452) [18,19]. There are four vibrational modes located at 92, 102, 121 and 141 cm^−1^ that can be observed in the Raman spectrum (Figure 2g), which can be assigned to the E_1_-TO, E_1_-LO, A_1_, and E_2_ peaks of Te, respectively [20,21,22]. The characteristic XRD peaks of Te NTs show up in Figure 2h, proving that the target product was successfully synthesized [23,24,25].

The photographs and SEM images of the Te-based electrode are illustrated in Figure 3. The surface of the electrode is smooth without any gaps or flaws, illustrating the structural integrity of the electrode. The photograph of the bent electrode in Figure 3b gives a good indication of the excellent flexibility of the Te-based electrode. Figure 3c shows that Te NTs on the surface of the electrode are not oriented and are stacked together. From the cross-section images of the Te-based electrode in Figure 3d, it can be observed that NFCs are interwoven to form a thin layer of flexible substrate (the red dotted region). Upon the NFCs substrate, Te NTs close-knit interlaced together and formed the active substance layer. Figure 3e is the high-resolution image of the yellow dotted box in Figure 3d, and numerous Te NTs can be clearly observed. It is evident that the NFCs and Te NTs are tightly compacted and agglutinated, and thus forming the flexible and freestanding electrode without the need for an additional binder or current collector.

The electrochemical performance of the flexible Te-based electrode is evaluated. Figure 4a exhibited the initial three CV curves of the Te-based electrode. The Te-based electrode shows two distinct peaks during the first negative sweep, which can be attributed to the forming of Li_2_Te and the solid electrolyte interphase (SEI) film, respectively. The corresponding reversible behaviors are clearly observed during the first charge process. The following reaction equation reveals the one-step electrochemical reaction mechanism of the Te NTs: Te +2 Li = Li_2_Te [13,26]. The CV curves of the following two cycles show similar shapes and near peak positions, and the peak intensity and integral area of the third cycle even increased than that of the second cycle, demonstrating the high reversibility and good electrochemical stability of the Te-based electrode. As shown in Figure 4b, the Te-based electrode delivers a high initial discharge capacity of 2219 mAh cm^−3^, while the corresponding initial Coulombic efficiency is only about 46%. The large irreversible capacity is probably related to subreaction between the electrolyte and electrode surface. Through comparing the charge–discharge curves for different cycles, it can be concluded that the volumetric capacity can maintain stability after several cycles. The capacity and coulomb efficiency of the Te-based electrode increase with cycling in the first several cycles. The phenomenon of an initial drop in capacity followed by a gradual increase is the so-called activation period for many nanoscale electrode materials. This phenomenon and the corresponding reasons have been found and verified by many previous studies [27,28]. The recovery of the capacity fading can be mainly attributed to the enhanced reaction kinetics and increased reactive sites of the Te NTs, which have been activated in the subsequent cycles. The good cycle stability can also be observed from the cycle performance in Figure 4c. With a current density of 200 mA g^−1^, the Te-based electrode delivers a discharge capacity of 1580 mAh cm^−3^ and a capacity retention of 104% after cycled 300 times. The high volumetric capacities of the flexible Te-based electrode can be mainly attributed to its huge density. The Te-based electrode also shows a good rate capability. As presented in Figure 4d, the Te-based electrode delivers a discharge capacity of 833 mAh cm^−3^ at a high current density of 1000 mA g^−1^, and returns back to 1195 mAh cm^−3^ at 200 mA g^−1^. The nanotube structure increase electrode–electrolyte contact area, which can favor the active phase accessibility and increase the utilization of the electroactive materials. The dominant volumetric capacity and excellent cycling stability of the Te-based electrode prove great potential in the applications for lithium batteries.

The SEM images of the Te-based electrode cycled 300 times are illustrated in Figure 5. The SEM image demonstrates that the surface of the cycled electrode is maintained as flat and intact. It is evident from Figure 5b that the cycled Te NTs are generally in good shape, which indicates the superior resistance to the volume change during the repeated cycles. Figure 5c,d show the TEM images of Te NTs cycled 300 times. Figure 5c shows that the Te NTs remain the original nanotube structure. Figure 5d presents the HRTEM image of the Te NT with a distinct crystal lattice. The lattice space of the Te NT is 0.39 nm, which is slightly larger than that of the freshly made Te NT shown in Figure 2f. The results suggest that the lattice space of the Te NTs will slightly increase with electrochemical cycling, and the nanotube structure of Te can suppress the repeated volume change in cycling, and thus maintaining the intact integrity of Te NTs and enabling the flexible Te-based electrode with achievements of durable cycling and high-rate capability.

In order to assess its applied prospect in flexible batteries, the cyclical bending test and the corresponding electrochemical measurement of the Te-based electrode are performed. As shown in Figure 6a, the Te-based electrode was repeatedly bent for 50 cycles with bending radii of 5 mm. The initial discharge capacity of the flexible electrode tends to degrade following bending cycles. After being bent 50 times, the Te-based electrode exhibits an initial discharge capacity of 816 mAh cm^−3^, and maintained 80% of its initial capacity. The cycle performance of the Te-based electrode after 50 bending cycles is depicted in Figure 6b. The bended Te-based electrode delivers a discharge capacity of 1039 mAh cm^−3^ After 100 cycles at 200 mA g^−1^, and the corresponding capacity retention rate is 93%. Given this superior deformation robustness, the Te-based electrode will pave the way towards flexible batteries.

## 4. Conclusions

In summary, uniform Te NTs were successfully synthesized and compounded with NFCs to form a flexible and free-standing Te-based composite electrode without extra conductive carbon and binder. The obtained Te-based electrode delivers a high volumetric capacity of 1512 mAh cm^−3^ when the current density is 200 mA g^−1^, and exhibits admirable cyclic stability (capacity retention of 104 % over 300 cycles) and high-rate capability (833 mAh cm^−3^ at 1000 mA g^−1^), which benefits from the unique structure and intrinsically superior conductivity of Te NTs. The Te-based electrode also showed significant durability through repeated deformations. The outstanding comprehensive properties of the Te-based electrode can be ascribed to the synergistic effect of Te NTs with excellent electrochemical performance and NFCs with superior mechanical properties.

## Figures and Tables

**Figure 1 nanomaterials-11-02903-f001:**
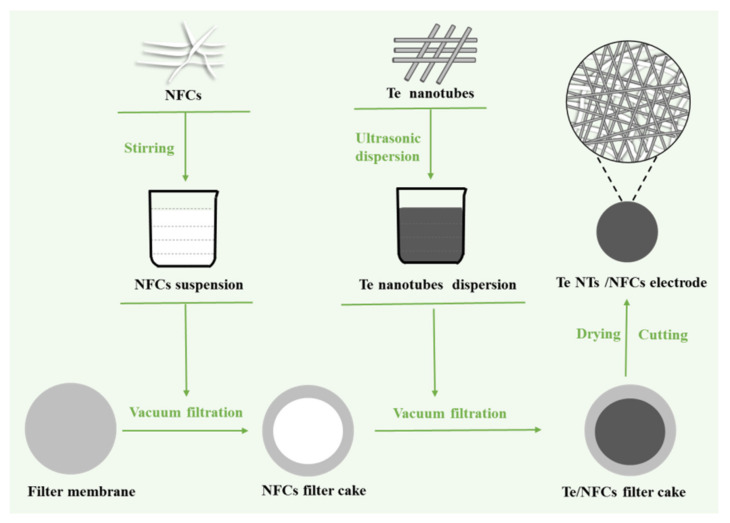
Flow diagram of the preparation procedure of flexible Te-based electrode.

**Figure 2 nanomaterials-11-02903-f002:**
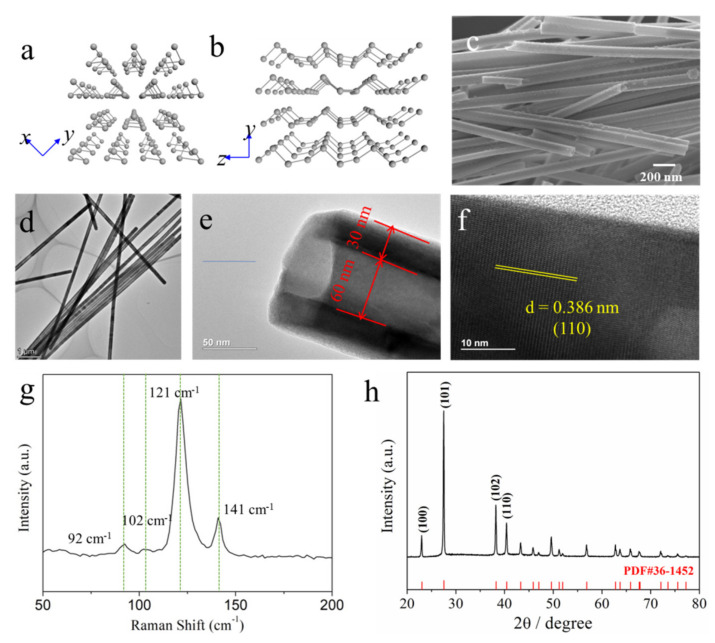
Characteristics of Te nanotubes: (**a**,**b**) Crystal structure; (**c**) SEM image; (**d**–**f**) TEM image; (**g**) Raman spectrum; (**h**) X-Ray Diffraction pattern.

**Figure 3 nanomaterials-11-02903-f003:**
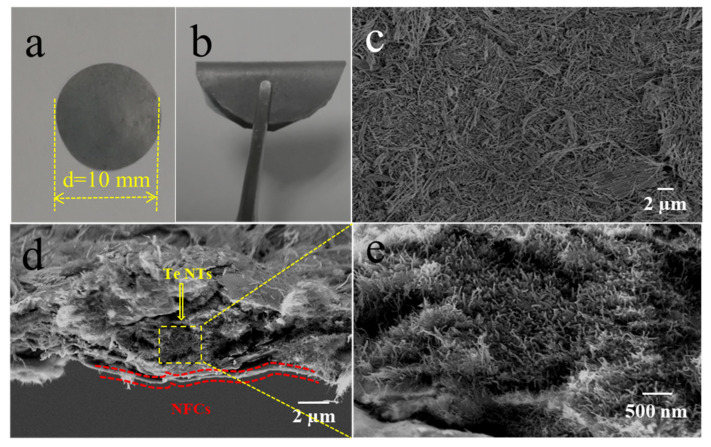
(**a**,**b**) The photos of the flexible Te-based electrode under flat and bent states; (**c**) SEM image of the Te-based electrode; (**d**,**e**) Cross-sectional SEM images of the Te-based electrode.

**Figure 4 nanomaterials-11-02903-f004:**
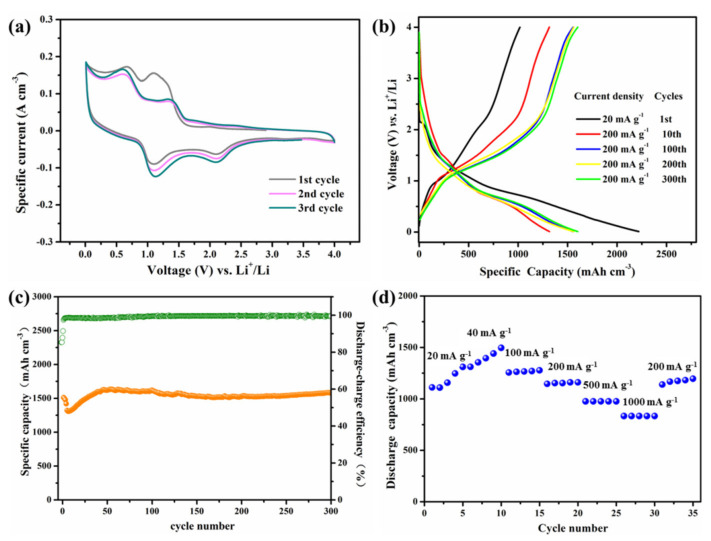
Electrochemical performance of the flexible Te-based electrode: (**a**) CV profiles for the initial three cycles; (**b**) Representative charge-discharge curves of the 1st, 10th, 100th, 200th and 300th cycles; (**c**) Cycle performance and Coulombic efficiency at 200 mA g^−1^ for 300 cycles and (**d**) Rate performance in the rate range of 20–1000 mA g^−1^.

**Figure 5 nanomaterials-11-02903-f005:**
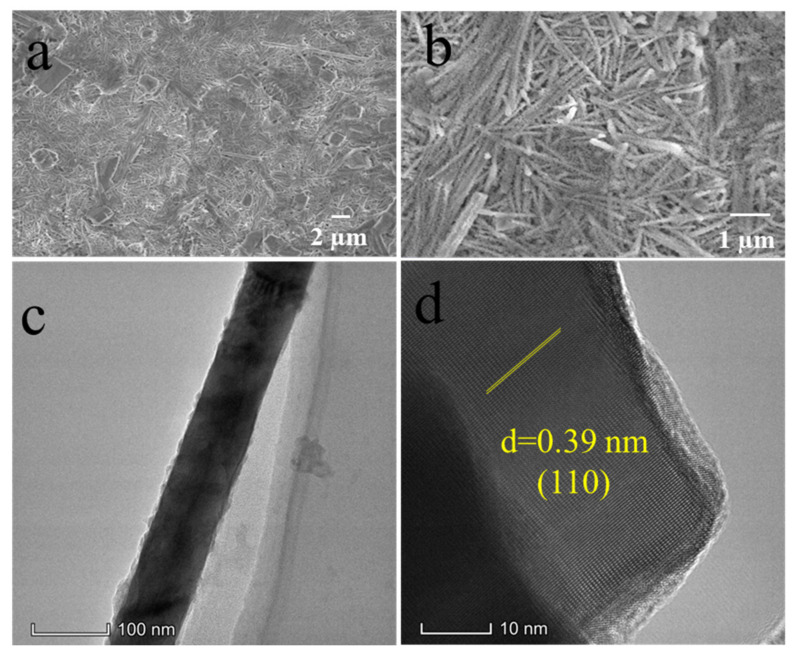
(**a**,**b**) SEM of the Te-based electrode; (**c**,**d**) TEM images of Te NTs after 300 cycles.

**Figure 6 nanomaterials-11-02903-f006:**
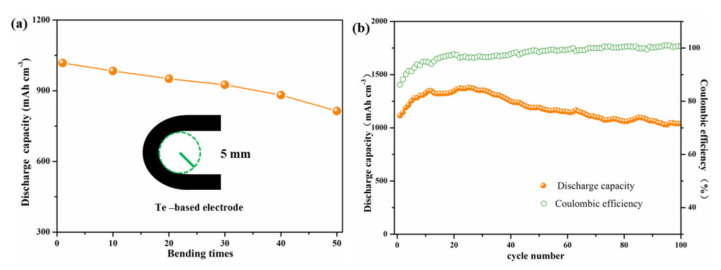
(**a**) The schematic of the bending test for the Te-based electrode, and its initial discharge capacity after different bending times; (**b**) Cycle performance and Coulombic efficiencies of the Te-based electrode after 50 bends to a radius of 5 mm.

## Data Availability

Not applicable.

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
