# Peer review of "Flexible Tellurium-Based Electrode for High-Performance Lithium-Tellurium Battery"

_nanomaterials, 2021, doi:10.3390/nano11112903_

Round 1
Reviewer 1 Report
Dear Author,
Your paper presents interesting data relative to the possible use of Composite Te NTs - NCFs electrodes associated with Li metal electrode for flexible batteries.
I have minor comments relative to the presentation/content of the paper and some scientific questions that I will list below, in order of appearance in the text:
- Lines 45-46: in the introduction part, it is indicated that the lack of flexibility of the conventional electrodes is limited by the rigid active materials. I think this should be better explained as the active materials are usually micrometric powders and the bendability of the electrodes are more limited by the adhesion/cohesion properties linked to the formulation of the electrode to my knowledge.
- Line 60: typo error in electrochemical
- Line 70: grammar error in "thus will resulting "
- Line 77: is expected to be a promising (be missing)
- Lines 93 to 99: I think this part should be better explained also. Especially, it would be nice to give the precise composition (%w of the different compounds of the composite electrode) and also to give the precise value of the active material loading (mg/cm²) and the coated thickness of the electrode (µm).
- Lines 102-106 (Materials description) but also everywhere in the text: I think that many abbreviations are not defined (if I have well read) such as CV, EIS, TEM, SEM... Even if it is something rather obvious, this should be included in the paper
- Lines 107-114 (Electrochemical testing); there is a lack of description for the tests: what is the counter electrode used (Li metal? which grade, thickness, provider, size), electrolyte chosen? separators?
- Line 125: that can be (that missing at the end of the line)
- Line 135: I would replace "demonstrates" by something weaker (gives a good indication for instance). I doubt that a picture of a bended electrode demonstrates something
- Line 149: typo error: which can be corresponded should be corrected (can be attributed or can correspond to ...)
- Line 156: 2219 cm-3 => I believe that 2219 mAh.cm-3 should be indicated
- Line 158: you mention a possible parasitic reaction between electrolyte and electrolyte surface to explain the large irreversible capacity. Aren't there any other possible mechanisms? (insulating products? the overall reaction mechanism during lithiation could be indicated somewhere in the article)
- Lines 165-167: the phrase is written twice. One should be removed
- An important global comment: I think that the indicated performances of the Te electrodes are obtained by cycling between 3V and 0V vs Li+/Li, but practically this is not possible for a battery. The voltage below 1V will be hardly exploitable by any electronic device, and so this should be mentionned somehow in the text.
Best regards
Reviewer 2 Report
This manuscript reports on the “Flexible Tellurium-based electrode for high-performance Lithium-Tellurium Battery”. The authors prepared Tellurium nanotubes using the hydrothermal method. The prepared electrode shows a high volumetric capacity of 1512 mAh cm-3 at 200 mA g-1, and 16 delivers admirable cyclic stability (capacity retention of 104 % over 300 cycles) and excellent rate 17 performance (833 mAh cm-3 at 1000 mA g-1). Overall, the presented results are interesting, however, there are several major issues with the manuscript which should be resolved before publication.
- A lot of work has already been done on Lithium-Tellurium batteries. The authors should clearly mention the novelty of this work in the introduction part.
- Complete experimental information is required for this manuscript. Such as final electrode condition, material loading, electrolyte, etc, are missing in the experimental part.
- It is recommended to present capacity values in mAh/g.
- What is the reason for increasing capacity after the initial 5 cycles? It is recommended to investigate the cell using EIS to get insights.
- Some powerful ex-situ analysis such as high-resolution TEM, XAS, etc, are suggested for the investigation of the mechanism.
Round 2
Reviewer 2 Report
The authors have addressed my comments and the manuscript can be accepted now.